# Relationships of Sexual Orientation Microaggression with Anxiety and Depression among Lesbian, Gay, and Bisexual Taiwanese Youth: Self-Identity Disturbance Mediates but Gender Does Not Moderate the Relationships

**DOI:** 10.3390/ijerph182412981

**Published:** 2021-12-09

**Authors:** Jung-Sheng Chen, Yu-Te Huang, Chung-Ying Lin, Cheng-Fang Yen, Mark D. Griffiths, Amir H. Pakpour

**Affiliations:** 1Department of Medical Research, E-Da Hospital, Kaohsiung 82445, Taiwan; nicky071214@gmail.com; 2Department of Social Work and Social Administration, The University of Hong Kong, Hong Kong RM543, China; yuhuang@hku.hk; 3Institute of Allied Health Sciences, College of Medicine, National Cheng Kung University, Tainan 70101, Taiwan; 4Department of Occupational Therapy, College of Medicine, National Cheng Kung University, Tainan 70101, Taiwan; 5Department of Public Health, College of Medicine, National Cheng Kung University, Tainan 70101, Taiwan; 6Biostatistics Consulting Center, National Cheng Kung University Hospital, College of Medicine, National Cheng Kung University, Tainan 70101, Taiwan; 7Department of Psychiatry, School of Medicine College of Medicine, Kaohsiung Medical University, Kaohsiung 80708, Taiwan; 8Department of Psychiatry, Kaohsiung Medical University Hospital, Kaohsiung 80756, Taiwan; 9College of Professional Studies, National Pingtung University of Science and Technology, Pingtung 91201, Taiwan; 10International Gaming Research Unit, Psychology Department, Nottingham Trent University, Nottingham NG1 4FQ, UK; mark.griffiths@ntu.ac.uk; 11Department of Nursing, School of Health and Welfare, Jönköping University, 55318 Jönköping, Sweden; Amir.Pakpour@ju.se

**Keywords:** microaggression, self-identity disturbance, anxiety, psychological well-being, sexual orientation

## Abstract

The aims of this cross-sectional survey study were to examine the association between sexual orientation microaggression and anxiety and depression among young adult lesbian, gay, and bisexual (LGB) individuals in Taiwan, as well as to examine the mediating effect of self-identity disturbance and the moderating effect of gender. In total, 1000 self-identified LGB individuals participated in the study. The experience of sexual orientation microaggression was assessed using the Sexual Orientation Microaggression Inventory, self-identity disturbance was assessed using the Self-Concept and Identity Measure, anxiety was assessed using the State subscale on the Chinese version of the State-Trait Anxiety Inventory, and depression was assessed using the Center for Epidemiological Studies-Depression Scale. Structural equation modeling (SEM) was used to examine relationships between the variables. The SEM results demonstrated that sexual orientation microaggression was directly associated with increased anxiety and depression, as well as being indirectly associated with increased anxiety and depression via the mediation of self-identity disturbance among young adult LGB individuals. Gender did not moderate the relationships between any of the variables. Both sexual orientation microaggression and self-identity disturbance warrant program interventions for enhancing mental health among LGB individuals.

## 1. Introduction

Lesbian, gay, and bisexual (LGB) individuals experience more severe anxiety and depression compared with their heterosexual peers [1,2]. Disparities in mental health among LGB individuals adversely affect their development in social relationships, as well as academic and occupational achievements [3]. Various forms of stigma at the individual level (e.g., identity concealment), interpersonal level (e.g., bullying victimization), and structural level (e.g., laws and social norms) increase the risk of adverse health outcomes among young LGB individuals [4,5,6]. In addition to overt acts of sexual prejudice, young LGB individuals may also experience covert acts of sexual prejudice, such as sexual orientation microaggression [7,8]. ‘Microaggression’ emerged as a term for describing acts of subtle racism and has been defined as “brief and commonplace daily verbal, behavioral, or environmental indignities that communicate hostile, derogatory, or negative racial slights or insults” [9] (p. 72). Sexual orientation microaggression is further classified into three forms: microassault, microinsult, and microinvalidation [8,9].

Microassaults refer to “overt verbal or nonverbal insults and behaviors against LGB individuals rooted in heterosexism” (for example, using heterosexist language such as “That’s so gay” to connote that something is bad or weird) [7,10]. Microinsults refer to “subtle statements or actions that may slight or demean LGB persons’ marginalized identity” (for example, making a joke that a gay man could not possibly like sports or commenting that a woman is “too pretty to be a lesbian”) [7,10]. Microinvalidations refer to “exclude, negate, or nullify the psychological thoughts, feelings, or experiential reality of LGB groups” (for example, LGB individuals receiving comments such as “Don’t be so sensitive” when talking about a stigmatized experience) [7,9,10].

Research has shown that the experience of sexual orientation microaggression increases the risks of depression and anxiety among LGB individuals [7,11,12,13,14]. Microaggression may result in feelings of being hurt, offended, or frustrated, and their mental health may be further compromised [10]. However, very few studies have examined the indirect mechanisms that may link microaggression to adjustment outcomes [15]. Hatzenbuehler [16] hypothesized that among sexual minorities, individuals’ cognitive processes, self-concepts, and coping mechanisms may mediate the relationship between prejudice events and mental health outcomes. Research has demonstrated that self-stigma [13], expected rejection [13], and rumination [13,17] mediate the relationship between sexual orientation microaggression and psychological distress among LGB individuals.

The present study examined the mediating effect of self-identity disturbance on the relationship between sexual orientation microaggression and anxiety and depression among LGB individuals. The development of self-identity is distinct from sexual orientation identity development. Sexual orientation is an independent component of a person’s sexual identity. Several studies have identified that genes and the prenatal hormone environment shape the development of the brain in humans; interactions between biological and cultural-environmental factors further influence the expression of sexual behaviors [18,19]. Regarding self-identity, identity formation is a crucial developmental task that begins in early childhood and is consolidated during young adulthood [20,21,22]. According to Erikson [21,22,23], normative identity development occurs when a person explores available opportunities and options and begins to make commitments to others and take on self-defining roles; those who achieve a consolidated identity experience a sense of consistency across time and contexts, and demonstrate stable attitudes, beliefs, and values. However, the processes of normative identity development may go awry and result in identity confusion. Several types of self-identity disturbance have been identified. Disturbed identity refers to “the inability to commit to typical roles and a tendency to acquire the thoughts, feelings, beliefs, and problems of others in adulthood” [24] (p. 645), (for example, “Sometimes I pick another person and try to be just like them, even when I’m alone” [25]). Unconsolidated identity refers to “the failure to make commitments to others, take on self-defining roles, and demonstrate stable beliefs, attitudes, and values” [26] (p. 33) (for example, “When someone describes me, I am not sure if they are right or wrong” [25]). Lack of identity refers to “the sudden and dramatic shifts in self-image with respect to goals, values, vocational aspirations, sexual identity, and types of friends” [25] (p. 234) (for example, “I feel like a puzzle and the pieces don’t fit together” [25]). Research found that victimization and homophobic bullying increased the risk of self-identity disturbance among gay and bisexual men in emerging adulthood [27]. Moreover, self-identity disturbance was found to be associated with mood disorders [28,29]. Whether self-identity disturbance mediates the relationship between sexual orientation microaggression and anxiety and depression among LGB individuals is unclear. Moreover, Hatzenbuehler [16] noted that the mediation of cognitive processes, self-concepts, and coping mechanisms for the relationship between prejudice events and mental health outcomes may be moderated by gender among LGB individuals. Therefore, whether or not gender moderates the mediation of self-identity disturbance warrants further study.

The present study has two aims. First, to examine the associations of sexual orientation microaggression with anxiety and depression as well as the mediating effect of self-identity disturbance on the association among young adult LGB individuals. Based on the results of previous studies [7,11,12,13,14], it was hypothesized that sexual orientation microaggression will be significantly associated with depression and anxiety among young adult LGB individuals (Hypothesis 1a). Given that victimization and homophobic bullying has been found to increase the risk of self-identity disturbance among gay and bisexual men in emerging adulthood [27] and that self-identity disturbance has been associated with mood problems [28,29], it was hypothesized that self-identity disturbance would mediate the associations of sexual orientation microaggression with anxiety and depression among young adult LGB individuals (Hypothesis 1b). Second, to examine the moderating effect of gender on the mediation of self-identity disturbance. A recent meta-analysis showed that the variability of the associations between microaggression and adjustment outcomes in studies was partially accounted for by gender [15]. Therefore, it was hypothesized that gender would moderate the associations of sexual orientation microaggression with anxiety and depression and the mediation of self-identity disturbance (Hypothesis 2).

## 2. Materials and Methods

### 2.1. Participants and Procedure

Participants were recruited by posting an online advertisement on social media, including Facebook, Twitter, and LINE (a direct messaging app), the Bulletin Board System, and the home pages of three health promotion and counseling centers for LGB individuals from August 2018 to July 2020. The inclusion criteria were individuals who identified their sexual orientation as being homosexual or bisexual, aged between 20 and 30 years, and living in Taiwan. Anyone who intended to participate in the present study could telephone the research assistants. The research assistant ensured the eligibility of potential participants for recruitment, explained the study aims and procedures to them, and scheduled the time for completing the study questionnaires with them individually in the study room. Ten potential participants were excluded due to the ineligibility of age (younger than 20 or older than 30). The research assistants interviewed the participants face-to-face in the study room to determine whether they had impaired intellect or any signs of alcohol and substance use that might interfere with understanding the study’s purpose and method or their ability to respond to the questions. If they had, they were excluded from the study. In accordance with the plan, we consecutively recruited 500 male and 500 female participants into this study and then stopped recruiting further participants. Informed consent was obtained from all participants prior to the assessment. The study was approved by the Institutional Review Board of Kaohsiung Medical University Hospital (KMUHIRB-F(II)-20180018).

### 2.2. Measures

#### 2.2.1. Sexual Orientation Microaggression Inventory (SOMI)

The 19-item traditional Chinese version [30] of the SOMI [8] was used to assess microaggression on the dimensions of anti-gay attitudes and expressions, denial of homosexuality, and societal disapproval over six months among LGB individuals. Psychometric evaluation has demonstrated the SOMI has a bifactor structure. There is a general factor in the SOMI that is separate from the four trait factors [8]. Items are rated on a five-point Likert type scale from 1 (not at all) to 5 (about every day). A higher SOMI score indicates a higher level of microaggression. The traditional Chinese version of the SOMI has acceptable internal consistency (Cronbach’s α = 0.90) and concurrent validity (correlations with familial stigma and psychological inflexibility: r = 0.336 and 0.262, respectively; *p* < 0.001) [30]. Cronbach’s alpha of the SOMI in the present study was 0.90.

#### 2.2.2. Self-Concept and Identity Measure (SCIM)

The traditional Chinese version of the 27-item SCIM was used to assess the level of current self-identity disturbance [25,31]. The SCIM assesses three dimensions of self-identity disturbance, including disturbed identity (for example, “Sometimes I pick another person and try to be just like them, even when I’m alone”), unconsolidated identity (for example, “When someone describes me, I am not sure if they are right or wrong”), and lack of identity (for example, “I feel like a puzzle and the pieces don’t fit together”). Items are rated on a seven-point rating scale ranging from 1 (strongly disagree) to 7 (strongly agree). A higher total score indicates a higher tendency for self-identity disturbance. The traditional Chinese version of the SCIM has acceptable congruent validity with bullying victimization [27] and predictive validity for depression and suicidality one year later [32]. Cronbach’s alpha of the SCIM in the present study was 0.79.

#### 2.2.3. State Subscale on the Chinese Version of the State-Trait Anxiety Inventory (STAI-S)

Twenty items from the self-administered Chinese version of the STAI-S were used to assess current anxiety symptoms (for example, “I feel tense”) [33,34]. Items are rated on a four-point Likert scale, with scores ranging from 1 (not at all) to 4 (very much so). Higher total STAI-S scores indicate more severe anxiety. The Chinese version of the STAI-S has acceptable test–retest reliability (Pearson’s r = 0.76), internal reliability (Cronbach’s α = 0.91), criterion validity (correlation with the Hamilton Anxiety Rating Scale: r = 0.69), and construct validity [35]. Cronbach’s alpha of the STAI-S in the present study was 0.89.

#### 2.2.4. Mandarin Chinese Version of the Center for Epidemiological Studies-Depression Scale (MC-CES-D)

The 20-item MC-CES-D was used to assess the frequency of depressive symptoms in the preceding month of the study (for example, “I was bothered by things that usually don’t bother me”) [36,37]. Items are rated on a four-point scale from 1 (rarely or none of the time) to 4 (most or all the time). Higher scores indicate more severe depression. The MC-CES-D has good internal consistency (Cronbach’s α = 0.90), one-week test–retest reliability (intraclass correlation reliability = 0.93), congruent validity (area under the receiver operative characteristic curves for major depressive disorder = 0.88–0.90) [38], and construct validity [39]. Cronbach’s alpha of the MC-CES-D in the present study was 0.93.

#### 2.2.5. Demographic and Sexual Orientation Factors

Data were collected regarding the participants’ gender, age, education level (high school or below vs. college or above), sexual orientation (homosexual or bisexual), and age of identification of sexual orientation (“When did you first identify yourself as gay or bisexual?”).

### 2.3. Data Analysis

The participants’ demographic information was first analyzed using descriptive statistics. Pearson correlations were then used to understand the associations between the studied variables in the proposed model for the entire sample, male sample, and female sample. Structural equation modeling (SEM) with the maximum likelihood estimator was used to examine the proposed model. In the SEM, the SATI-S and MC-CES-D total scores were used to represent the observed variables of anxiety and depression, respectively. The SOMI domain scores were used to construct latent sexual orientation microaggression. The SCIM domain scores were used to construct latent self-identity disturbance. Fit indices, including comparative fit index (CFI), Tucker–Lewis index (TLI), root mean square error of approximation (RMSEA), and standardized root mean square residual (SRMR) were applied to examine whether the collected data fits with the proposed model. A good fit should have CFI and TLI > 0.9 together with RMSEA and SRMR < 0.08 [40]. After confirming the fit of the proposed model, multigroup SEM was used to examine whether gender was a moderator in the proposed model. More specifically, χ^2^ difference tests were used to examine whether a constrained path was significantly different from a freely-estimated path. When the significance is found, gender is a significant moderator in the path. SEM and multigroup SEM were analyzed using the R software with the lavaan package [41]. All other analyses were performed using the IBM SPSS 20.0 (IBM Corp., Armonk, NY, USA).

## 3. Results

Table 1 shows the study sample’s characteristics (*N* = 1000; 500 males and 500 females). In brief, the mean age of the participants was 24.63 years (SD = 2.99) and the mean age at which the participants identified their sexual orientation was 14.48 years (SD = 3.86). Most of the participants were well educated (89.1% had a college or above educational level), and over half were homosexual (57.0%).

The correlations between the studied variables were all significant, except for the confounder of age (Table 2). Moreover, the correlation magnitudes were similar across gender. The proposed model was fully supported by the fit indices (CFI = 0.967, TLI = 0.948, RMSEA = 0.068, and SRMR = 0.039). Furthermore, all the path coefficients were significant (*p* < 0.001), with the strongest path coefficient being between self-identity disturbance and depression (standardized coefficient = 0.703), followed by the path between self-identity disturbance and anxiety (standardized coefficient = 0.558) (Figure 1).

Multigroup SEM showed that all the path coefficients were significant for male and female participants (Figure 2). Sexual orientation microaggression was directly associated with increased anxiety and depression, as well as indirectly associated with increased anxiety and depression via the mediation of self-identity disturbance. The fit indices of the multigroup SEM also indicated that the proposed model was supported by the data separated by gender. Moreover, χ^2^ difference tests showed that the path coefficients were not significantly different between male and female participants: χ^2^ = 2.87 (df = 1; *p* = 0.09) for sexual orientation microaggression to anxiety; χ^2^ = 0.48 (df = 1; *p* = 0.49) for self-identity disturbance to anxiety; χ^2^ = 0.29 (df = 1; *p* = 0.59) for sexual orientation microaggression to depression; χ^2^ = 0.14 (df = 1; *p* = 0.71) for self-identity disturbance to depression; and χ^2^ = 0.01 (df = 1; *p* = 0.92) for sexual orientation microaggression to self-identity disturbance. The results indicated that gender did not moderate the associations of sexual orientation microaggression with anxiety and depression or the mediation of self-identity disturbance.

## 4. Discussion

The results of the present study demonstrated that sexual orientation microaggression was directly associated with increased anxiety and depression, as well as being indirectly associated with increased anxiety and depression via the mediation of self-identity disturbance among young adult LGB individuals. The results supported both Hypotheses 1a and 1b. However, gender did not moderate the associations of sexual orientation microaggression with anxiety and depression or the mediation of self-identity disturbance. Therefore, the results did not support Hypothesis 2.

Sexual orientation microaggression is less overt compared with bullying. However, it may damage the victims’ psychological well-being in several ways. Firstly, sexual orientation microaggression may make LGB individuals feel misunderstood by (and different from) others, and the feeling of loneliness may deteriorate LGB individuals’ mental health [42]. Secondly, enactors of microaggression might view their own behavior as harmless, unremarkable, or well-intentioned [10]. Targets of microaggressions may feel helpless or powerless due to the difficulties in communicating and reaching consensus with the enactors regarding the bias shown in microaggression. The sense of helplessness and powerlessness may further compromise individuals’ mood regulation [43]. Thirdly, LGB individuals may delay the timing of seeking medical advice for mood problems for fear of experiencing sexual orientation microaggression [44,45].

The present study found that sexual orientation microaggression was also indirectly associated with increased anxiety and depression via the mediation of self-identity disturbance among young adult LGB individuals. Research has found that victims may experience a clash of realities arising from sexual orientation microaggression [10,46,47]. The disagreement between the victims’ perspectives and perpetrators’ perspectives toward “sexual orientation microaggression” may make some LGB individuals confused about the “reality” in which they live. Given that self-identity is the result of interaction between individuals and environments in the ecological view [48], a clash of realities arising from the experience of sexual orientation microaggression may disturb the formation of self-identity. Furthermore, sexual orientation microaggression may evoke the ‘catch-22’ of responding among LGB individuals [10,46,47]. Sexual orientation microaggression may form a paradoxical situation from which the victims cannot escape because of multiple facts, such as diversity of perspectives, repercussions for confronting the perpetrators, and lack of energy and time to engage in such conversations [10]. The paradoxical situations resulting from sexual orientation microaggression may interfere with the efforts of LGB individuals to develop stable beliefs, attitudes, and values. Self-identity disturbance may further increase the risk of mood dysregulation [28,29]. Because of the cross-sectional study design, the possibility that LGB individuals who had self-identity disturbance were more vulnerable to sexual orientation microaggression should be taken into account. However, sexual orientation microaggression is prevalent and exists in the daily lives of people, regardless of self-identity or mental health.

Hatzenbuehler [16] suggested that gender may moderate the mediation of self-concepts for the relationship between prejudice events and mental health outcomes. There are gender differences in the developing process of self-identity [49] and mood problems [50]. Research has also demonstrated that gender may partially account for the variability in the results of previous studies on the links between microaggression and adjustment outcomes [15]. However, the present study did not support the moderating effect of gender on the relationships between sexual orientation microaggression, self-identity disturbance, or mood problems among LGB individuals.

The present study confirmed again the significant association between sexual orientation microaggression and anxiety and depression among young adult LGB individuals; moreover, this study is the first one to identify the mediating effect of self-identity disturbance on the association between sexual orientation microaggression and anxiety and depression among LGB individuals. The results support that both efforts to change sexual orientation microaggression in the social dimension and to improve self-identity disturbance in the individual dimension are needed for the mental health of LGB individuals. Based on the results of the present study, a number of proposals have been formulated. First, given that sexual orientation microaggression is a common experience among LGB individuals, it is necessary to help LGB individuals develop adequate strategies to cognitively and emotionally cope with the experience of microaggression, and also to successfully communicate with the perpetrators. Second, research has suggested that public health strategies addressing attitudes to sexual orientation and promoting the changes of subtle microaggression among the general population may contribute to diverse affirmative cultural scripts regarding LGB individuals’ lives and enhancing mental health [51,52]. Research has demonstrated the effects of the programs for enhancing cultural competency to interact with sexual minorities among medical professionals [53] and students [54]. Research has also suggested integrating the roles and ability of the active bystanders into the intervention programs for reducing microaggressions [55]. Third, positive self-identity development could be promoted, especially for LGB individuals experiencing sexual orientation microaggression. For example, based on identity status theory [56], clarifying individuals’ identity through fostering exploration may enhance identity commitment [57] and helps individuals become more mature and competent during life transitions [58].

### Limitations

There are some limitations in the present study. First, the cross-sectional study design limited the inferences concerning the temporal relationships between sexual orientation microaggression, self-identity disturbance, anxiety, and depression. For example, self-identity disturbance, anxiety, and depression might increase LGB individuals’ sensitivity to sexual orientation microaggression. Therefore, further prospective studies are needed to examine the temporal relationships between these variables. Second, the present sample comprised young adults (aged between 20 and 30 years) who were well-educated (nearly 90% of the participants had a college degree or above). Therefore, it is unclear whether the results of the present study could be generalized to the populations with other age ranges or with a lower level of education. Third, all the data collected in the present study were self-reported. Therefore, single-rater biases, recall biases, and social desirability biases cannot be fully controlled [59,60,61]. Fourth, state anxiety is the individual’s emotional arousal and unpleasant feelings to the situation, which can change along with time and circumstances. The state anxiety measured in this study indicated the participant’s anxiety while completing the interview and might not be generalized to other situations. Fifth, this study inquired participants’ gender identities by the binary of man and woman but did not include the options of transgender, gender nonbinary, or genderqueer. Research has found that sexual and gender minority identities have intersectional impacts on health [62] and behaviors [63]; both sexual and gender minority identities should be considered in public health practice [64].

## 5. Conclusions

The present study found that sexual orientation microaggression was directly associated with anxiety and depression, as well as indirectly associated with increased anxiety and depression via the mediation of self-identity disturbance among young adult LGB individuals. Consequently, mental health professionals should routinely assess the experience of sexual orientation microaggression among LGB individuals who suffer from anxiety and depression and help them develop effective strategies to cope with microaggression. Self-identity disturbance should also be assessed and interventions provided to prevent emotional problems among LGB individuals.

## Figures and Tables

**Figure 1 ijerph-18-12981-f001:**
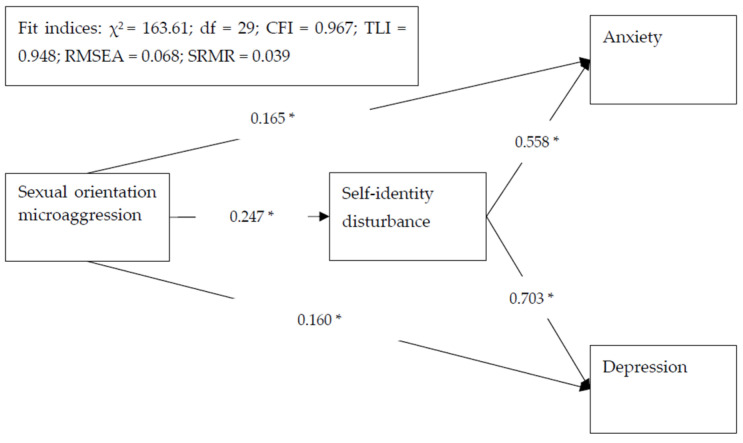
Structural equation model results with standardized regression coefficients for the proposed model. Age was controlled in the model. * *p* < 0.001.

**Figure 2 ijerph-18-12981-f002:**
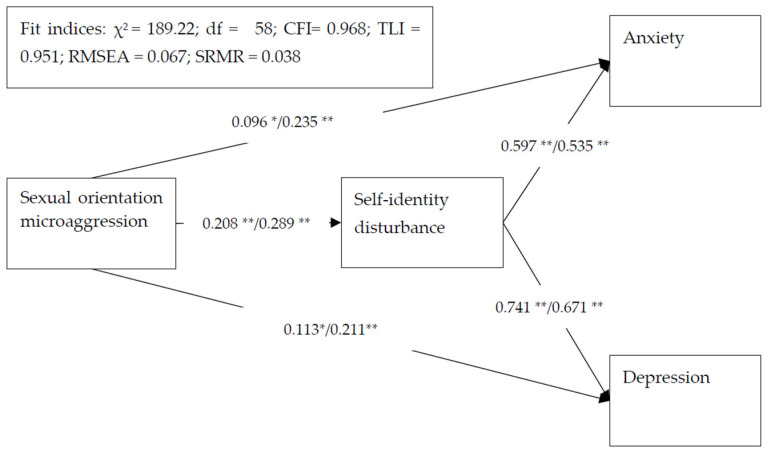
Multigroup structural equation model results with standardized regression coefficients for the proposed model. Age was controlled in the model. Standardized regression coefficients presented for the male sample/female sample; no moderated effects in sex were observed (*p*-values of χ^2^ difference tests = 0.09 to 0.92). * *p* < 0.05; ** *p* < 0.001.

**Table 1 ijerph-18-12981-t001:** Participants’ characteristics (*N* = 1000).

	Total n (%)	Males n (%)	Females n (%)
Age (years) ^a^	24.63 (2.99)	24.80 (2.91)	24.45 (3.06)
Age of identifying sexual orientation (years) ^a^	14.48 (3.86)	13.96 (3.77)	15.00 (3.89)
Gender			
Male	500 (50.0)	-	-
Female	500 (50.0)	-	-
Educational level			
High school or below	109 (10.9)	64 (12.8)	45 (9.0)
College or above	891 (89.1)	436 (87.2)	455 (91.0)
Sexual orientation			
Bisexual	430 (43.0)	135 (27)	295 (59)
Homosexual	570 (57.0)	365 (73)	205 (41)

^a^ Reported using mean (SD).

**Table 2 ijerph-18-12981-t002:** Correlation matrix between studied variables.

	Mean (SD)	Range				*r* (*p*-Value)		
**Total sample**			1.	2.	3.	4.	5.	6.
1. Age (years)	24.63 (2.99)	20–30	--					
2. Age of identifying sexual orientation (years)	14.48 (3.86)	5–29	0.01 (0.79)	--				
3. Sexual orientation microaggression	42.01 (11.56)	19–79	0.02 (0.59)	−0.08 (0.01)	--			
4. Self-identity disturbance	96.17 (24.16)	33–186	−0.08 (0.02)	0.03 (0.35)	0.20 (<0.001)	--		
5. Anxiety	40.82 (12.67)	20–79	0.02 (0.51)	0.05 (0.13)	0.27 (<0.001)	0.54 (<0.001)	--	
6. Depression	18.79 (11.22)	0–57	−0.02 (0.52)	0.01 (0.67)	0.30 (<0.001)	0.66 (<0.001)	0.73 (<0.001)	--
**Male sample**			1.	2.	3.	4.	5.	6.
1. Age (years)	24.80 (2.91)	20–30	--					
2. Age of identifying sexual orientation (years)	13.96 (3.77)	5–26	0.07 (0.17)	--				
3. Sexual orientation microaggression	43.24 (11.24)	19–79	0.06 (0.19)	−0.07 (0.14)	--			
4. Self-identity disturbance	98.09 (25.11)	33–174	−0.02 (0.62)	0.05 (0.26)	0.17 (<0.001)	--		
5. Anxiety	40.19 (12.60)	20–79	0.02 (0.62)	0.01 (0.89)	0.19 (<0.001)	0.55 (<0.001)	--	
6. Depression	18.44 (11.42)	0–57	0.04 (0.40)	−0.04 (0.32)	0.23 (<0.001)	0.69 (<0.001)	0.74 (<0.001)	--
**Female sample**			1.	2.	3.	4.	5.	6.
1. Age (years)	24.45 (3.06)	20–30	--					
2. Age of identifying sexual orientation (years)	15.00 (3.89)	5–29	−0.03 (0.58)	--				
3. Sexual orientation microaggression	40.77 (11.76)	19–79	−0.03 (0.46)	−0.07 (0.14)	--			
4. Self-identity disturbance	94.25 (23.03)	41–186	−0.15 (0.001)	0.04 (0.38)	0.21 (<0.001)	--		
5. Anxiety	41.44 (12.71)	20–79	0.03 (0.57)	0.77 (0.09)	0.35 (<0.001)	0.55 (<0.001)	--	
6. Depression	19.14 (11.02)	0–57	−0.07 (0.10)	0.02 (0.70)	0.37 (<0.001)	0.64 (<0.001)	0.73 (<0.001)	--

## Data Availability

Other researchers can access the data for independent verification under reasonable request to the corresponding authors (Lin, C.-Y. and Yen, C.-F.).

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
