# Peer review of "Relationships of Sexual Orientation Microaggression with Anxiety and Depression among Lesbian, Gay, and Bisexual Taiwanese Youth: Self-Identity Disturbance Mediates but Gender Does Not Moderate the Relationships"

_ijerph, 2021, doi:10.3390/ijerph182412981_

Round 1

Reviewer 1 Report

This study presents the results of an SEM mediation and moderation analysis of cross-sectional interview data collected among LGB young adults. The results show that self-reported experiences of microaggressions is related to self-identity disturbance, which is also related to state-level anxiety and past month depressive symptoms. There were no significant differences in these relations between men and women in the sample. The significance of the study could be emphasized. The authors could add additional detail describing how this study moves the field forward. 

Abstract – the first sentence could be broken up into two sentences, as currently written it is a little bit hard to follow.

Introduction - The paragraph about self-identity disturbance could be expanded to include examples of self-identity disturbance as well as normative self-identity development. The paragraph currently includes many quoted definitions to explain the construct but lacks specific examples. The paragraph could also explain how self-identity disturbance is distinct from sexual orientation identity development.

Methods - The manuscript states that no participants were excluded, yet the study includes exactly 500 men and 500 women. This seems unlikely that the sample would have a perfect 50/50 split across sample demographics without some oversight by the researchers. Can the authors please report the number of participants screened, then of those, who were eligible and included in the study? Additionally, were transgender participants excluded?

Self-Concept and Identity measure – the authors call this construct self-identity confusion in the methods, but call this self-identity disturbance in the introduction. Please stay consistent throughout the manuscript.

STAI-S –The state anxiety questionnaire may be measuring someone’s anxiety while completing the interview, rather than generalized levels of anxiety, whereas the measure of depression used measure depressive symptoms over the last month. Did the authors also include the trait anxiety measure from the STAI? Why was the state questionnaire used?

Table 1 – would be nice to see crosstabs of demographics by gender.

Author Response

We appreciated your valuable comments. As discussed below, we have revised our manuscript with underlines based on your suggestions. Please let us know if we need to provide anything else regarding this revision.

Comment 1

The significance of the study could be emphasized. The authors could add additional detail describing how this study moves the field forward. 

Response

Thank you for your suggestion. We added describing how this study moves the field forward. Please refer to line 309-315.

“The present study confirmed again the significant association between sexual orientation microaggression and anxiety and depression among young adult LGB individuals; moreover, this study is the first one to identify the mediating effect of self-identity disturbance on association between sexual orientation microaggression and anxiety and depression among LGB individuals. The results supported that both efforts to change sexual orientation microaggression in the social dimension and to improve self-identity disturbance in the individual dimension are needed for mental health of LGB individuals.”

Comment 2

Abstract – the first sentence could be broken up into two sentences, as currently written it is a little bit hard to follow.

Response

Thank you for your suggestion. We revised this sentence as below. Please refer to line 29-32.

“The aims of this cross-sectional survey study were to examine the association between sexual orientation microaggression and anxiety and depression among young adult lesbian, gay, and bisexual (LGB) individuals in Taiwan, as well as to examine the mediating effect of self-identity disturbance and moderating effect of gender.”

Comment 3

Introduction

  1. The paragraph about self-identity disturbance could be expanded to include examples of self-identity disturbance as well as normative self-identity development. The paragraph currently includes many quoted definitions to explain the construct but lacks specific examples.
  2. The paragraph could also explain how self-identity disturbance is distinct from sexual orientation identity development.

Response

Thank you for comments.

  1. We added the sample for normative self-identity development and self-identity disturbance as below.

“According to Erikson [21-23], normative identity development occurs when a person explores available opportunities and options, and begins to make commitments to others and take on self-defining roles; those who achieve a consolidated identity experience a sense of consistency across time and contexts, and demonstrate stable attitudes, beliefs, and values.” Please refer to line 90-94.

“Disturbed identity...(for example, “Sometimes I pick another person and try to be just like them, even when I’m alone” [25]). Unconsolidated identity...(for example, “When someone describes me, I am not sure if they are right or wrong” [25]). Lack of identity ...(for example, “I feel like a puzzle and the pieces don’t fit together” [25]).” Please refer to line 98-105.

  1. We added the contents as below into the revised manuscript to introduce how self-identity disturbance is distinct from sexual orientation identity development. Please refer to line 83-88.

 “The development of self-identity is distinct from sexual orientation identity development. Sexual orientation is an independent component of a person's sexual identity. Several studies have identified that gene and prenatal hormone environment shape the development of the brain in humans; interactions between biological and cultural-environmental factors further influence the expression of sexual behaviors [18,19].

Comment 4

Methods

  1. The manuscript states that no participants were excluded, yet the study includes exactly 500 men and 500 women. This seems unlikely that the sample would have a perfect 50/50 split across sample demographics without some oversight by the researchers. Can the authors please report the number of participants screened, then of those, who were eligible and included in the study?
  2. Additionally, were transgender participants excluded?

Response

Thank you for your comments.

  1. This study planned to recruit 500 male and 500 female individuals who identified their sexual orientation as being homosexual or bisexual, aged between 20 and 30 years, and living in Taiwan. Anyone who intended to participate in the present study could telephone the research assistants. The research assistant ensured the eligibility of potential participants for recruitment. In total, 10 potential participants were excluded due to the ineligibility of age (younger than 20 or older than 30). We stopped recruiting further participants after successfully recruiting 500 male and 500 female participants. We added these explanations as below into the revised manuscript. Please refer to line 143-144 and 148-149.

“Ten potential participants were excluded due to the ineligibility of age (younger than 20 or older than 30)...In accordance with the plan, we consecutively recruited 500 male and 500 female participants into this study and then stopped recruiting further participants.”

  1. We did not exclude transgender participants from this study. Transgender participants labelled their gender as what they considered themselves were. However, we agreed that gender minority identities should be included into consideration. We listed it as one of the limitations of this study as below. Please refer to line 349-353.

“This study inquired participants’ gender identities by the binary of man and woman but did not include the options of transgender, gender nonbinary, or genderqueer. Research has found that sexual and gender minority identities have intersectional impacts on health [62] and behaviors [63]; both sexual and gender minority identities should be considered in public health practice [64].

62. Hsieh, N.; Ruther, M. Sexual minority health and health risk factors: Intersection effects of gender, race, and sexual identity. Am. J. Prev. Med. 2016, 50, 746-755. doi: 10.1016/j.amepre.2015.11.016.

  1. Lewis, B.J.; Hesse, C.L.; Cook, B.C.; Pedersen, C.L. Sexistential crisis: An intersectional analysis of gender expression and sexual orientation in masculine overcompensation. J. Homosex. 2020, 67, 58-78. doi: 10.1080/00918369.2018.1525943.
  2. Sell, R.L.; Krims, E.I. Structural transphobia, homophobia, and biphobia in public health practice: The example of COVID-19 surveillance. Am. J. Public Health 2021, 111, 1620-1626. doi: 10.2105/AJPH.2021.306277.

Comment 5

Self-Concept and Identity measure – the authors call this construct self-identity confusion in the methods, but call this self-identity disturbance in the introduction. Please stay consistent throughout the manuscript.

Response

Thank you for your reminding. We unified them into “self-identity disturbance”. Please refer to line 171 and 177.

Comment 6

STAI-S –The state anxiety questionnaire may be measuring someone’s anxiety while completing the interview, rather than generalized levels of anxiety, whereas the measure of depression used measure depressive symptoms over the last month. Did the authors also include the trait anxiety measure from the STAI? Why was the state questionnaire used?

Response

Thank you for your comment. We have considered using the anxiety-state inventory or the anxiety-strait inventory. Trait anxiety refers to individual’s relatively stable response to threat and stress, is usually related to personality, and reflects individual differences. Because that the present study examined the association of sexual orientation microaggression with anxiety and the mediation of self-identity disturbance, examining trait anxiety may increase the difficulty in clarifying the relationships among the variables. Therefore, we measured state anxiety in this study. However, state anxiety is the individual’s emotional arousal and unpleasant feelings to the situation which can change along with time and circumstances. The state anxiety measured in this study indicated the participant’s anxiety while completing the interview and might not be generalized to other situations. We added it as one of the limitations of this study as below. Please refer to line 345-349.

“State anxiety is the individual’s emotional arousal and unpleasant feelings to the situation which can change along with time and circumstances. The state anxiety measured in this study indicated the participant’s anxiety while completing the interview and might not be generalized to other situations.”

Comment 7

Table 1 – would be nice to see crosstabs of demographics by gender.

Response

We added demographics by gender into Table 1. Please refer to line 230.

Reviewer 2 Report

  1. The minority stress model has been criticized, see Bailey et al. 2020.  One of the major limitations is that cross sectional data are used to infer causal direction in only one way.  For example, stigma might reduce mental health but lower mental health might be a cause of stigma.
  2. I am not sure about the need or intent associated with the phrase "intentional or unintentional" on line 61.  For centuries, intention has played an important role in deciding how bad an offense was and the seeming equation of intentional with unintentional looks like a step backwards in justice to this reviewer.
  3. Is it not possible that self-identity disturbance might elicit bullying?  Bullies are usually cowards who like to pick on those weaker or confused.
  4. Likewise, if you are anxious might not that lead to greater sensitivity to possible microaggressions? 
  5. Line 142 onward.  Please provide means and SDs and ranges for the key measures.  It is being argued that the scores will be higher but without knowing the averages, it's hard to say.  The range for SOMI could be 19 to 95 but the average might be 25.6 or it might be 87.3, which would represent huge differences.  The averages, means, and ranges must be reported.
  6. The time frames for the four measures should be reported.  The time frame is for some but not for all at present.
  7. The age of identification of sexual orientation should be included in the correlation matrix.  The ranges of the variables in Table 1 should be reported as well except for the obvious ones that have only two levels.
  8. Line 260, the period should be a comma?
  9. Line 342, it is not clear if the meaning was that the other co-authors have not seen the data (often associated with fraud) or if the authors mean other researchers could have the data for independent verification.
  10. It would be helpful to give a couple of items as examples for each of the multiple item scales for readers who may not have used those scales.

Author Response

Comment 1

The minority stress model has been criticized, see Bailey et al. 2020. 

Response

Thank you for your reminding. We deleted the sentence “The minority stress hypothesis suggests that stigmatizing attitudes and behaviors marginalizes LGB individuals and negatively affects their mental and physical health [4].” from the manuscript. Please refer to line 51.

Comment 2

One of the major limitations is that cross sectional data are used to infer causal direction in only one way.  For example, stigma might reduce mental health but lower mental health might be a cause of stigma.

Response

We agree your comment. In 4.1. Limitation section, we added the description as below to emphasize the limitation of the cross-sectional study design in this study. Please refer to line 336-338.

“First, the cross-sectional study design limited the inferences concerning the temporal relationships between sexual orientation microaggression, self-identity disturbance, anxiety and depression. For example, self-identity disturbance, anxiety and depression might increase LGB individuals’ sensitivity to sexual orientation microaggression.

Comment 3

I am not sure about the need or intent associated with the phrase "intentional or unintentional" on line 61.  For centuries, intention has played an important role in deciding how bad an offense was and the seeming equation of intentional with unintentional looks like a step backwards in justice to this reviewer.

Response

Thank you for your suggestion. We deleted “...whether intentional or unintentional...” from the manuscript. Please refer to line 58.

Comment 4

Is it not possible that self-identity disturbance might elicit bullying?  Bullies are usually cowards who like to pick on those weaker or confused. Likewise, if you are anxious might not that lead to greater sensitivity to possible microaggressions? 

Response

We agree your comment. In 4.1. Limitation section, we added the description as below to emphasize the limitation of the cross-sectional study design in this study. Please refer to line 336-338.

“First, the cross-sectional study design limited the inferences concerning the temporal relationships between sexual orientation microaggression, self-identity disturbance, anxiety and depression. For example, self-identity disturbance, anxiety and depression might increase LGB individuals’ sensitivity to sexual orientation microaggression.

Comment 5

Line 142 onward.  Please provide means and SDs and ranges for the key measures.  It is being argued that the scores will be higher but without knowing the averages, it's hard to say.  The range for SOMI could be 19 to 95 but the average might be 25.6 or it might be 87.3, which would represent huge differences.  The averages, means, and ranges must be reported.

Response

Thank you for reminding. We added the means and SDs and ranges for age, age of identifying sexual orientation, sexual orientation microaggression, self-identity disturbance, anxiety and depression into Table 2. Please refer to line 239.

Comment 6

The time frames for the four measures should be reported.  The time frame is for some but not for all at present.

Response

We added the time frames as below for the four measures.

2.2.1. Sexual Orientation Microaggression Inventory (SOMI)

“...in the six months...” Please refer to line 161.

2.2.2. Self-Concept and Identity Measure (SCIM)

“...current... Please refer to line 171.

2.2.3. State subscale on the Chinese version of the State-Trait Anxiety Inventory (STAI-S)

“...current... Please refer to line 183.

2.2.4. Mandarin Chinese version of the Center for Epidemiological Studies-Depression Scale (MC-CES-D)

...the preceding month of the study.” Please refer to line 192.

Comment 7

The age of identification of sexual orientation should be included in the correlation matrix.  The ranges of the variables in Table 1 should be reported as well except for the obvious ones that have only two levels.

Response

Thank you for reminding. We added the age of identifying sexual orientation into Table 2. We also added means and SDs and ranges for the key variable into Table 2. Please refer to line 239.

Comment 8

Line 260, the period should be a comma?

Response

Thank you for your reminding. We corrected it. Please refer to line 278.

Comment 9

Line 342, it is not clear if the meaning was that the other co-authors have not seen the data (often associated with fraud) or if the authors mean other researchers could have the data for independent verification.

Response

We revised this announcement into “Other researchers could have the data for independent verification under the reasonable request to the corresponding authors (Lin, C.-Y. and Yen, C.-F.).” Please refer to line 377-378.

Comment 10

It would be helpful to give a couple of items as examples for each of the multiple item scales for readers who may not have used those scales.

Response

Thank you for your suggestion. We added the example items as below into Methods section.

2.2.1. Sexual Orientation Microaggression Inventory (SOMI)

“The 19-item traditional Chinese version [28] of the SOMI [9] was used to assess microaggression on the dimensions of anti-gay attitudes and expressions (for example, “You heard someone say “that’s so gay” in a negative way?”), denial of homosexuality (for example, “You were told you just haven’t found the right person of the opposite sex”), heterosexualism (for example, “You were told you’re not a “real” man”), and societal disapproval (for example, “Someone said a hateful slur about gay, lesbian, and bisexual people”) among LGB individuals.” Please refer to line 156-161.

2.2.2. Self-Concept and Identity Measure (SCIM)

The SCIM assesses three dimensions of self-identity disturbance, including disturbed identity (for example, “Sometimes I pick another person and try to be just like them, even when I’m alone”), unconsolidated identity (for example, “When someone describes me, I am not sure if they are right or wrong”), and lack of identity (for example, “I feel like a puzzle and the pieces don’t fit together”).” Please refer to line 172-176.

2.2.3. State subscale on the Chinese version of the State-Trait Anxiety Inventory (STAI-S)

“Twenty items from the self-administered Chinese version of the STAI-S were used to assess current anxiety symptoms (for example, “I feel tense”).” Please refer to line 183.

2.2.4. Mandarin Chinese version of the Center for Epidemiological Studies-Depression Scale (MC-CES-D)

“The 20-item MC-CES-D was used to assess the frequency of depressive symptoms in the preceding month of the study (for example, “I was bothered by things that usually don't bother me”).” Please refer to line 192-193.

Round 2

Reviewer 2 Report

It was great to provide examples of the items used but for one scale you repeat the examples, giving them twice, which isn't necessary.

Author Response

Dear reviewer:

Thank you for your suggestion. We deleted the example items for the Sexual Orientation Microaggression Inventory in 2.2.1. section.